# A Preliminary Study of the Influence of Graphene Nanoplatelet Specific Surface Area on the Interlaminar Fracture Properties of Carbon Fiber/Epoxy Composites

**DOI:** 10.3390/polym12123060

**Published:** 2020-12-21

**Authors:** Konstantina Zafeiropoulou, Christina Kostagiannakopoulou, George Sotiriadis, Vassilis Kostopoulos

**Affiliations:** 1Department of Mechanical Engineering & Aeronautics, University Campus Patras, GR-26504 Rio Achaia, Greece; k_zafeiropo@upnet.gr (K.Z.); kostagia@mech.upatras.gr (C.K.); sotiriad@mech.upatras.gr (G.S.); 2Foundation of Research and Technology, Institute of Chemical Engineering Sciences (FORTH/ICE-HT), Stadiou Str., GR-26504 Rio Achaia, Greece

**Keywords:** fracture properties, graphene nanoplatelets, specific surface area, composite laminates, SEM

## Abstract

Graphene nanoplatelets (GNPs) are of particular interest to the field of nano-reinforced composites since they possess superior mechanical, fracture, thermal, and barrier properties. Due to their geometrical characteristics, high aspect ratio (AR)/specific surface area (SSA) and their planar structure, GNPs are considered as high-potential nanosized fillers for improving performance of composites. The present study investigates the effect of SSA of GNPs on fracture properties of carbon fiber reinforced polymers (CFRPs). For this reason, two nano-doped CFRPs were produced by using two types of GNPs (C300 and C500) with different SSAs, 300 and 500 m^2^/g, respectively. Both types of GNPs, at the same content of 0.5 wt%, were added into the epoxy matrix of composites by applying a three-roll milling technique. The nanomodified matrix was used for the manufacturing of prepregs, while the final composite laminates were fabricated through the vacuum-bag method. Mode I and II interlaminar fracture tests were carried out to determine the interlaminar fracture toughness G_IC_ and G_IIC_ of the composites, respectively. According to the results, the toughening effect of C500 GNPs was the strongest, resulting in increases of 25% in G_IC_ and 33% in G_IIC_ compared with the corresponding unmodified composites. The activation of the absorption mechanisms of C500 contributed to this outcome, which was confirmed by the scanning electron microscopy (SEM) analyses conducted in the fracture surfaces of specimens. On the other hand, C300 GNPs, due to disability to be dispersed uniformly into the epoxy matrix, did not influence the fracture properties of CFRPs, indicating that probably there is a threshold in SSA which is necessary to achieve for improving the fracture properties of CFRPs.

## 1. Introduction

Carbon fiber reinforced polymers (CFRPs) are of great interest for aircraft structures, automotive, electronics, and infrastructure applications, due to their high specific stiffness and strength. In the view of their cross-linked structure, epoxy resins are used as matrices for CFRPs, owing to their high modulus, strength, and chemical/thermal resistance. However, they also come with the undesirable property of brittleness with low fracture toughness. Therefore, appropriate modifications for improving the fracture performance of these polymers have been developed by using nano-sized particles [1,2,3]. Carbon-based nanofillers, such as carbon nanofibers [4], carbon nanotubes (CNTs) [5,6,7,8,9], graphene nano-species (GNSs), or/and graphene nanoplatelets (GNPs) [10,11,12,13,14,15,16] are being widely investigated as reinforcements of epoxy resins.

Carbon-based nanofillers of a range of lateral size, thickness, aspect ratio, and surface functionality have been used to modify epoxy polymer, affecting differently the properties of the final composites. Hsieh et al. [17] studied the size effect of multi-walled carbon nanotubes (MWCNTs) on mechanical properties and toughening performance of polymer-based nanocomposites. Two types of nanotubes were used to manufacture the nanotube/epoxy nanocomposites: the standard, with long length (140 μm) and small diameter (120 nm), and the low-quality, with average length of 32 μm and diameter of 180 nm. Different weight percentages (0.1, 0.2, and 0.5 wt%) of both nanotubes (standard and low-quality) were added into the epoxy resin and dispersed using the sonication method. The results revealed that the maximum value of Young’s modulus was measured for the 0.5 wt% standard nanotube/epoxy sample, an increase of 20% compared to the unmodified epoxy polymer. The tensile strength exhibited a slight increase of 6% for 0.2 wt% standard CNTs. Finally, the addition of 0.5 wt% of the standard nanotubes yielded increases of 49 and 90% in fracture toughness and fracture energy, respectively, compared to the unmodified polymer. Furthermore, Zhang et al. [18] contributed to the investigation of the impact of the morphology and the structure of graphene nanosheets (GNSs) on the mechanical properties of GNS/epoxy nanocomposites. They fabricated GNS/epoxy nanocomposites with GNSs, having three different specific surface areas of 121 m^2^/g, 436 m^2^/g, and 487 m^2^/g (GNS0, GNS5, and GNS15, respectively). The weight fraction of GNSs in the epoxy matrix was held constant at 0.1 wt%. The tensile strength of the GNS15/epoxy nanocomposite was about 25.3% larger than that of the pristine epoxy. GNS0/epoxy and GNS5/epoxy composites showed slight increase in the tensile strength compared to the neat epoxy matrix. The modulus enhancements for GNS0, GNS5, and GNS15 composites at the same weight fraction of 0.1 wt% slightly increased. From the flexural tests, were observed that GNS15/epoxy composites prominently increased the flexural strength, which was about 30.9% larger than that of pristine epoxy. The strength enhancements for GNS0 and GNS5 reinforced composites at the same weight fraction of 0.1 wt% were also significantly increased. Finally, the modulus enhancements for all GNS composites at the same weight fraction of 0.1 wt% gradually increased. Chong et al. [19] was a member of another team that reported the influence of aspect ratio and surface functionality on mechanical properties and toughening mechanisms of GNPs modified epoxies. In total, six types of graphene nanoplatelets modifier were used: C-750, H-5, M-25, GS, GNP-COOH, and GNP-O2 with aspect ratios 19, 103, 1142, 205, 85, and 83, respectively. The non-functionalized GNPs were sonicated in tetrahydrofuran (THF) or *n*-methyl-pyrrolidone (NMP) to facilitate dispersion in epoxy resin. It was found that with the larger platelet size, and hence aspect ratio, GNPs gave higher values of modulus and Kc, except from M-25, which did not disperse well into the epoxy. The maximum values of modulus (at 1 wt%) and fracture energy (at 2 wt%) were measured for the epoxy modified with an intermediate platelet size of approximately 4 μm (205 aspect ratio), increasing the corresponding properties of the unmodified epoxy 24 and 257% respectively. Ravindran and his colleagues [20] investigated experimentally the affection of size and specific surface area of multilayer graphene nanoplatelets (GNPs) in the electrical properties of epoxy nanocomposites. GNPs used in manufacturing were at two different grades: grade C, with average particle diameter less than 2 μm and surface areas 300, 500, and 750 m^2^/g, and grade H, with average surface area 60–80 m^2^/g and particle diameter of 5, 15, and 25 μm. For assessing the influence of surface areas on the properties, GNPs C-300, C-500, and C-750 were employed at weight fraction 5 wt% and 10 wt%, respectively, while the addition of GNPs H-5, H-15, and H-25 into epoxy was carried out at concentrations of 1, 5, and 10 wt% GNPs, respectively, following the three-roll-milling process for the homogenization of nanofillers into the epoxy system. It was found that increasing the surface area and the concentration of the GNPs, results in an improvement in AC electrical conductivity, reaching 104% for 10 wt% C-750 nanotubes. As regards to the size effect, for all GNP sizes, no significant enhancement has been found in the AC electrical conductivity at 1 wt%. However, significant improvement (103–105%) is found for nanocomposites with 5 wt% and 10 wt% of GNPs. Lastly, the electrical conductivity increased dramatically (102%), when GNP diameter increased from 5 µm to larger size, while limited discrepancy was observed between the results with diameters being 15 and 25 µm, respectively.

As it is perceived from the above-mentioned studies, graphene can act as a multifunctional nanofiller into polymer matrices. Especially, graphene nano-species are predicted to have remarkable performance, such as high thermal conductivity, excellent electronic transport properties, and superior mechanical properties [21,22,23,24,25], which carries higher levels of transferring stress across the interface and provides higher reinforcement than carbon nanotubes (CNTs) [26]. Towards this direction, graphene modified CFRPs have attracted widespread interest in studying their mechanical and fracture properties. Leopold et al. [27] analyzed the impact of few-layered graphene (FLG) matrix modification in either 0°- or 90° layers of cross-ply laminates on the mechanical properties. The FLGs were dispersed in the resin by using a three-roll mill at a filler content of 0.3 wt%. The 0° layer modified (GNG) CFRPs managed to increase the Young’s modulus 5.8%, while the tensile strength remained almost unchanged compared the unmodified cross-ply composites. On the other hand, the 90° layer modified (NGN) CFRPs showed much higher value for tensile strength, increasing the difference with the unmodified specimens to 15.4%. Srivastava et al. [28] investigated Mode I and Mode II fracture toughness of woven carbon fabric reinforced epoxy resin composites that had been modified with 3 wt% GNPs. The proper dispersion of GNPs with the resin included mixing with a rod and mechanical stirring. The results from the fracture tests showed improved toughness GIC and GIIC than neat CFRP composites, scoring increases of 153% and 43%, respectively.

From the above literature, we can reach two conclusions. Firstly, it is obvious that the morphology of the nanofillers in the polymer matrix can influence the properties of the final composite and secondly, graphene nanofillers are very promising additives to enhance the mechanical and fracture performance of CFRPs. In this paper, trying to combine these two conclusions, we investigated the effect of different specific surface areas of GNPs on the fracture (Mode I and II) properties of CFRP composites.

## 2. Materials and Methods

### 2.1. Materials

A four part epoxy prepreg system provided by Huntsman Advanced Materials (Basel, Switzerland) was used as the matrix material. The typical characteristics of each part of the system is cited in Table 1, while, as recommended by the manufacturer, the components are typically mixed in the ratio of 100:23:5:12 by weight. The main reinforcement phase was a unidirectional non-crimp carbon fabric, 300 mm wide T700SC, supplied by TORAYCA (Tokyo, Japan) with an areal density of 200 g/m^2^ and a 75 dtex PET binding yarn on both sides.

As regards to the nano-fillers, graphene nanoplatelets (GNPs) were obtained in powder form from XG-Sciences (Lansing, MI, USA). Specifically, two types of GNPs grade C (C300 and C500) were used, altering their specific surface area. It is worth to be noted that the letter C states the grade of the graphene, whereas the followed number states the value of the SSA of the GNPs in m^2^/g. The physical characteristics of each nano-filler are listed in Table 2.

### 2.2. Preparation of the Nano-Modified Matrix

The dispersion of the graphene platelets in the epoxy resin was achieved using a three-roll milling process which is described analytically from Kostagiannakopoulou et al. [29]. Firstly, an appropriate amount of GNPs and epoxy resin were mixed inside a glove box by hand stirring for 2 min before milling. The glove box offers safety during the dispersion process, as the graphene nanoplatelets are dangerous when they are free in the atmosphere. Then, the mixture was fed through the feed roll and was collected at the apron roll. This process was carried out four times, while the speed of the apron roll was maintained at 270 rpm. Following this, the other three parts of the resin system were added to the prepared suspension and mixed manually for 2 min. The developed nano-modified mixture was finally degassed in a vacuum chamber in order to remove the air bubbles. The weight concentration of GNPs was constant at 0.5 wt% in the epoxy-based nanocomposites produced as referred previously.

### 2.3. Prepreg Manufacturing Process

Carbon fiber unidirectional prepregs were used in order to manufacture the final composite laminates. The prepreg technique followed was an automatic process through a set-up, which was designed and in-house built for this reason, as shown in Figure 1. In this automatic prepreger, the carbon fabric was initially wrapped in a fiber spool, while at the start of the process, was led to a resin bath, which included the appropriate volume of the resin system (neat or nanomodified) for the total impregnation. The temperature of the mixture was controlled through a cooling–heating system adjusted in the bath in order to achieve the desirable viscosity for the uniform heat transfer and distribution of the mixture in the carbon fabric. Then, the impregnated fabric, as had exited the resin bath, passed through two rollers, to which was applied pressure through pistons for the removal of the excess resin from the surface of the carbon fabric. Next, it was covered by two release papers, which in turn were wrapped in different drums from each other, contributing to the protection of the impregnated fabric. Later, the prepreg was driven again between two corresponding rollers, determining the final amount of the polymeric system included in the prepreg and consequently its final thickness. Finally, the prepreg cloth was wrapped in a drum winder, where it was left in room conditions for 48 h to semi-polymerize and reach the so-called B-stage, according to the manufacturer instructions. After two days, the prepreg fabric was stored at −15°C. In total, three different prepreg fabrics were manufactured including the neat resin system, nanomodified with C300 GNPs and nanomodified with C500 GNPs, respectively.

### 2.4. Preparation of Nano-Reinforced CFRP Laminates

To produce unidirectional CFRP laminates, 300 × 300 mm^2^ sheets were cut from the corresponding manufactured prepreg fabric. Each laminate was composed of 16 plies prepreg sheets and a polytetrafluoroethylene (PTFE) film, according the test standard, was placed in the middle plane of each laminate to generate the initial delaminated region. After completion of the lamination process, the plates were employed with vacuum bagging and then were cured in an autoclave oven according to the manufacturer instructions (120 °C for 2 h) and 6 bars pressure. For the fabrication of the unmodified CFRP, the same process was followed without the intermediate step of the three roll-milling process, since the matrix consisted of the bulk epoxy resin only. The fiber volume fractions (*v_f_*) of the manufactured laminates are cited in Table 3 and were calculated from the cured ply thickness (CPT) of each plate:(1)$$v_{f}=\frac{1}{1+\frac{w_{m}\times \rho_{f}}{w_{f}\times \rho_{m}}}$$
where *w_m_* = mass of matrix, *ρ_m_* = density of cured matrix, *w_f_* = mass of fibers, and *ρ_f_* = density of fibers.

Quality control for all the composites was performed utilizing the C-scan ultrasonic technique. The result of the C-scan method depicts the energy (amplitude) of the received signal from the second passage of the wave from the laminate (of the wave reflected in the glass) at the x-y level corresponding to the scan coordinates. The color gradient (left scale in Figure 2) corresponds to the energy-amplitude measurements with a low rate of return in green and full rate in red. This means that in the red areas, the receiver takes all the amount of energy sent, so no energy is lost or dissipated during propagation. Therefore, in these areas, we have no imperfections or damage. On the contrary, in areas with shades of green, the energy losses are large, and this in turn is interpreted as the presence of damage. The signal sent is attenuated due to the presence of interfaces (due to delamination), resulting in lower energy carrying material. All intermediate shades correspond to intermediate percentages of returned energy. The white areas indicate the presence of air or the absence of material.

The results of the C-scan showed satisfactory and acceptable quality without major defects (heterogeneities, porosity, thickness variations, and delamination), see Figure 2a–c. The white areas indicate the presence of the PTFE film.

## 3. Fracture Tests

### 3.1. Mode I Test

Double cantilever beam (DCB) tests were carried out to measure the opening Mode I interlaminar fracture toughness energy (GIC). For this reason, DCB specimens were cut by a ribbon from the manufactured laminates and five specimens were tested for each type of material system according to the AITM 1.0005 standard. The test configurations are shown in Figure 3. Loading blocks and specimens were grit blasted with sand paper before bonding and cleaned with acetone impregnated soft paper. Aluminum piano hinges were glued to the surfaces of the beam at the notch. One side of the specimens was painted white in order for the propagation of crack to be clearly visual. The DCB tests were performed in a servo-hydraulic test machine (Instron 8872) and the cross-head speed was set at 10 mm/min. The applied loading and the opening displacement of each specimen were recorded to measure the interlaminar fracture toughness energy GIC, while the location of the crack tip was tracked down in regular intervals and was recorded along with the load and the displacement at each measured crack extension. Mode I interlaminar fracture toughness energy GIC was calculated according to the following equation:(2)$$G_{IC}=\frac{A}{a+w}\times 10^{6}$$
where *A* = energy to achieve the total propagated crack length, *α* = propagated crack length, and *w* = specimen width.

### 3.2. Mode II Test

Three-point end-notched flexure (ENF) tests were carried out based on the procedure described in the AITM 1.0006 standard. The ENF specimens used 100 mm span length, 25 mm width, and 120 mm length with initial crack length of 35 mm. A schematic view of the three-point bending test is shown in Figure 4. Five specimens for each material system were tested and the experiments were done with the cross-head velocity of 1 mm/min. The Mode II interlaminar fracture toughness GIIC is calculated from the following equation:(3)$$G_{IIC}=\frac{9Pa^{2}d\times 100}{2w\left(\frac{1}{4}L^{3}+3a^{3}\right)}\times 10^{6}$$
where *P* = critical load to start the crack, *w* = specimen width, *L* = span length, *d* = crosshead displacement at crack delamination onset, and *α* = initial crack length.

## 4. Results and Discussion

### 4.1. Fracture Properties Mode I

The representative load versus displacement curves from the DCB tests of the neat and the nano-modified composites are shown in Figure 5. As noticed in the figure, all the curves have shown a linear increase of load with displacement, until it reaches a maximum load, corresponding to the point of crack initiation. It is observed that nano-doped composites have higher maximum load than the un-doped, offering more resistance against the start of crack initiation in CFRPs. Specifically, increasing the SSA of GNPs, a corresponding increment is seen in the value of maximum load, as the surface area of GNPs can act as desirable interface for stress transfer [30]. This linear elastic region is followed by a sudden decrease in the load, indicating the initiation of crack propagation from the starter delamination inserted in the mid-plane of the manufactured composites through the PTFE film. After this, it is obvious from Figure 5 that the crack propagates further with a ‘’stick-slip’’ way, a characteristic crack propagation pattern involved in the composites. When compared with the neat composite, the C500 nano-modified composite is presented with higher load values, whereas C300 nano-reinforced composite seems degraded. Subsequently, the incorporation of GNPs can positively influence the fracture performance of CFRPs, only if the SSA exceeds a potential threshold value.

The presence of a potential threshold value of SSA of GNPs is speculated from Table 4, in which interlaminar fracture toughness of all materials is represented. The material with the C500 nano-modified matrix managed to increase the interlaminar fracture toughness GIC 25% compared to the baseline material. On the other hand, for the C300/CFRP composite, interlaminar fracture toughness was lower than the neat CFRP. This difference is due to the alternative behavior of C300 and C500 GNPs in crack initiation and crack propagation. The crack initiation is mainly related to the matrix toughness, which is enhanced from the addition of GNPs into the resin matrix. That is why both GNPs correspond well to the crack initiation. However, propagation reflects the interaction between the epoxy matrix and the fiber reinforcement, giving a safer result for the total behavior of the composite in the fracture. C500 composites managed to react better to propagation due to the more effective activation of the filler absorption mechanism in this region (see Section 4.3). However, seems that C300 were unable to enhance the matrix-fiber interface, resulting in degraded value of fracture toughness.

### 4.2. Fracture Properties Mode II

ENF tests were conducted to measure the displacement at the central loading point, other than the crack opening displacement. Representative load-displacement curves of neat and nano-modified CFRP composites are shown in Figure 6. It is assumed that the point of deviation from linearity indicates the crack initiation. As can be seen from the Figure 6, the maximum load of the GNPs-modified composites are higher than the unmodified and specifically increasing the surface area of GNPs, it is observed a corresponding increment in the value of maximum load, due to the ability of GNPs acting as loading transfer, as mentioned above.

In ENF tests, only the initial values of the Mode II interlaminar fracture toughness GIIC can be measured. As can be seen from Table 5, the introduction of C300 GNPs into the matrix increased the fracture performance in Mode II only by 5% compared to the neat CFRP. On the other hand, similar to GIC values, the incorporation of C500 nano-fillers in the composite laminate has also shown a significant improvement in the Mode II fracture toughness. Specifically, an addition of 0.5 wt% of C500 improved the GIIC value from 1.02 to 1.36 kJ/m^2^ by 33%.

### 4.3. Fractographic Analysis

Failure mechanisms under DCB tests, at the level of mid-plane, were characterized using scanning electronic microscopy (SEM) analyses. The samples were attached to aluminum bar and were coated with gold to ensure the transmission of electrons. Figure 7 shows the SEM micrographs of the fracture surfaces of the baseline and the GNPs-modified CFRPs after the Mode I test. As it is illustrated in Figure 7a, the fracture surface of composites with pure epoxy is smooth, while the crack propagation follows the direction of fiber reinforcement and takes place in the fiber-matrix interface, revealing a weak adhesive bond between the fiber and the matrix. However, the fracture surfaces are rougher than the neat composite due to the existence of GNPs (Figure 7c–f). Furthermore, the fracture surfaces of GNPs-doped CFRPs consist of several small and different height fracture surfaces, while narrow bands were observed at the boundary of the developed fracture surfaces, which run parallel to the crack growth direction. These individual fracture surfaces emanate from the additional failure mechanisms introduced by the incorporation of graphene nanoplatelets during the crack propagation process.

As it is mentioned in other research reports, the failure mechanisms that take place in graphene-reinforced polymers and composites are categorized in three failure modes, (a) crack pinning by the nano-filler, (b) separation between the graphitic layers, and (c) shear failure due to difference in height on fracture surfaces [16,20,26,31,32]. In Figure 8 and Figure 9, the failure mechanisms that were observed in the doped composites during the study of their fracture surface are depicted. The first failure mode discovered in the fracture surfaces was the crack pinning as seen in the representative SEM images Figure 8a and Figure 9a,b. As the crack continues to propagate in the vicinity of GNPs, parts of the crack are pinned by the GNPs and the rest continue to propagate and wrap around them, bifurcating in two or more individual cracks. These individual cracks may connect again later or may connect with other cracks that exist at different height levels on the fracture plane. The circles in the SEM images indicate the GNPs that caused the separation of a crack, while the arrows, the crack branching. The second failure mechanism revealed from the incorporation of GNPs in fracture toughening is the pull-out (Figure 8b and Figure 9c,d). This mode occurs when the crack encounters the surface of the graphene sheets. The van der Waals forces between the sheets become secondary, so it is easier for the crack to continue to propagate between the graphene sheets and finally passes through by separating the graphitic layers. As a result of this failure mode, the sheets gradually slip away from the matrix and finally pull out, see Figure 8b and Figure 9c,d.

### 4.4. Effect of Specific Surface Area on the Fracture Performance

The addition of any kind of nanofiller increases the toughness of matrix material, as long as a proper connection with the matrix and a sufficient adhesion are ensured. An exploitation of the theoretical surface area of the nanofiller as interface to the epoxy matrix is related to the dispersion and the matrix impregnation. Thus, the interface is playing a major role concerning toughening of nanocomposites. It is well known that the geometry of the fillers is one of the most important parameters in determining both the dispersion state of fillers and consequently the fracture properties of CFRP composites. The nano-size of GNPs and the van der Waals attraction between the platelets increase the viscosity and limit the partial motion of polymer chains and, consequently, the macromolecular movement of the nano-reinforced matrix. Increasing the SSA of GNPs, the GNP interfiller spacing is reduced as argued by Noh et al. [33], as a result the presence of the polymer existing between the graphene platelets, which is the main cause for crack initiation in the composites, being reduced. Consequently, the higher SSA of GNPs reduces the possibility of crack propagation in the matrix. Moreover, the larger surface area of the GNPs may allow for a greater degree of exfoliation of the stacked graphene sheets during the three-roll milling process, leading to a more effective interaction between the nano-fillers [20].

Owing the lower SSA, C300 graphene platelets have a corresponding higher thickness of polymer matrix between the nano-fillers, enhancing the crack initiation and propagation. Furthermore, agglomerates inside the C300-modified composites were observed during the scanning electronic microscopy (SEM) process, see Figure 10, as the van der Waals forces were so dominant and kept close to the graphene sheets. The restacking is a phenomenon that occurs frequently during mixing with the polymer matrix due to strong van der Waals forces between the graphene fillers and causes cracks, pores, and pin holes in the composite. This can be correlated to the poor state of dispersion of C300 inside the matrix, leading to the conclusion that the second mechanism of exfoliation did not activate during the mixing process. In large agglomeration zones (bigger than 10 μm), the stress concentration regions constitute a further reason for decreasing the fracture toughness. The sizes of the agglomerates were measured for C300-modified composites from the fracture surfaces, and the agglomerates were about 9.5 μm in size as illustrated in Figure 10. This reduces the effective aspect ratio and hence has a negative effect on the fracture properties of the final CFRP composite.

## 5. Conclusions

GNPs with two different specific surface areas, 300 and 500 m^2^/g, were used as reinforcement in carbon fiber/epoxy composites and their interlaminar fracture toughness were investigated. Carbon fiber unidirectional prepregs were used in order to manufacture the multiscale laminated composites utilizing an in-house prepreger. The additional reinforcement of the matrix was applied by a calendering process, three roll mill. Double cantilever beam (DCB) and three-point end-notched flexure (ENF) tests were conducted to examine the fracture energy Mode I and Mode II, respectively. The fracture analysis of composites showed a remarkable improvement of 25% in GIC and 33% in GIIC for loading 0.5 wt% of C500 GNPs. The source of this enhancement can be raised from the better adhesion between the nanoparticles and the CFRP and the additional failure mechanisms introduced by the incorporation of graphene nanoplatelets during the crack propagation process, leading to a stronger load transfer between the reinforcement agents inside the matrix and the nano-enhanced composite which is more resistant to failure. While higher SSA of the GNPs showed significant increases in fracture properties of nano-enhanced composites, lower SSA of GNPs resulted in less improvements. Owing the lower SSA, C300 graphene platelets have a corresponding higher thickness of polymer matrix between the nano-fillers, strengthening the crack initiation and propagation. In combination with the poor state of dispersion of GNPs and the forming of agglomerates inside the matrix (SEM), the effective aspect ratio reduced and hence, the fracture properties of the CFRP composite were negatively affected.

## Figures and Tables

**Figure 1 polymers-12-03060-f001:**
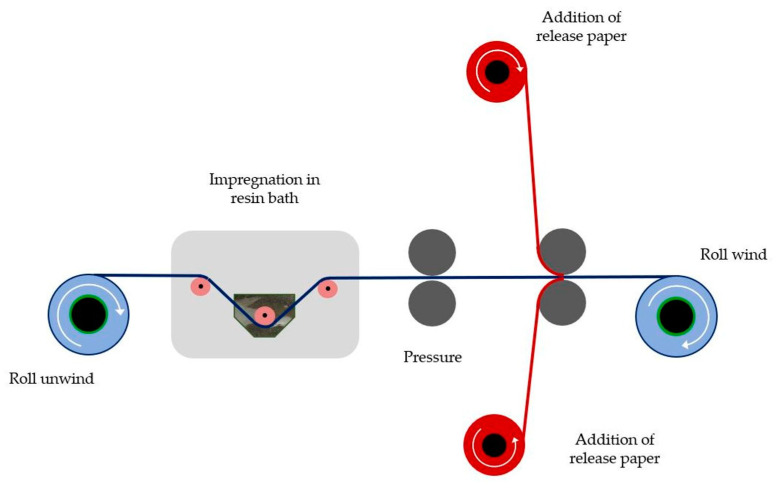
Schematic presentation of fabrication of prepregs with an in-house technique.

**Figure 2 polymers-12-03060-f002:**
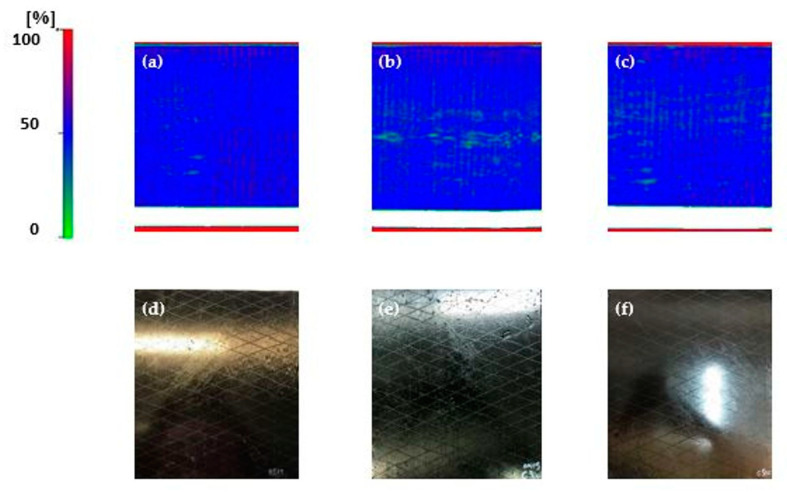
C-scan representation of the manufactured composites, (**a**) neat, (**b**) C300, and (**c**) C500, and the illustration of the manufactured composites, (**d**) neat, (**e**) C300, and (**f**) C500.

**Figure 3 polymers-12-03060-f003:**
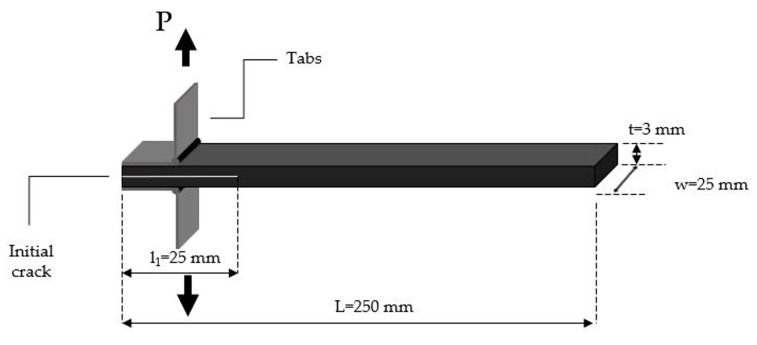
Specimen geometry for fracture toughness Mode I test.

**Figure 4 polymers-12-03060-f004:**
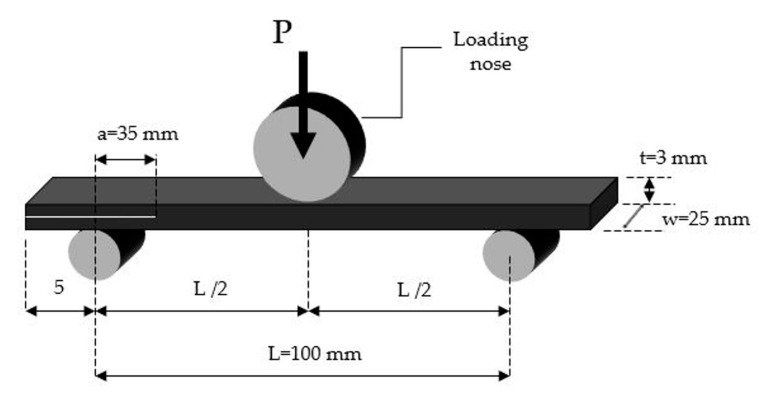
Specimen geometry for fracture toughness Mode II test.

**Figure 5 polymers-12-03060-f005:**
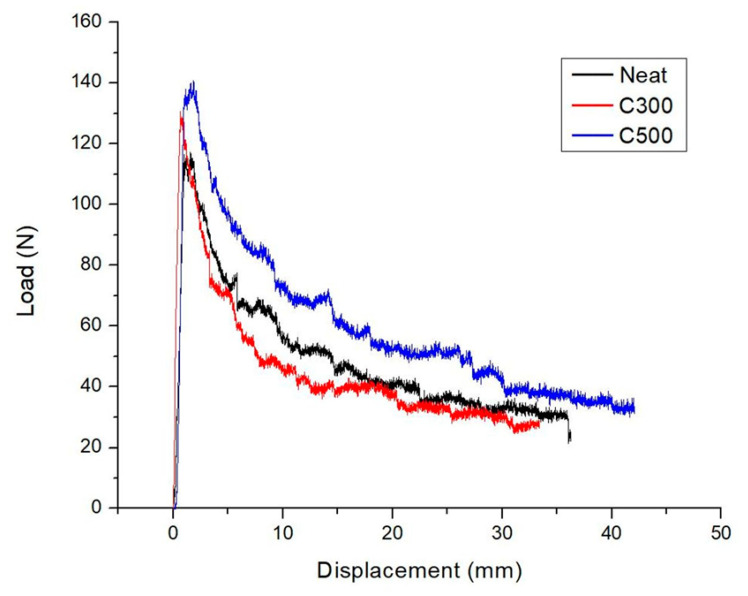
Load-displacement curves of representative specimens under DCB tests.

**Figure 6 polymers-12-03060-f006:**
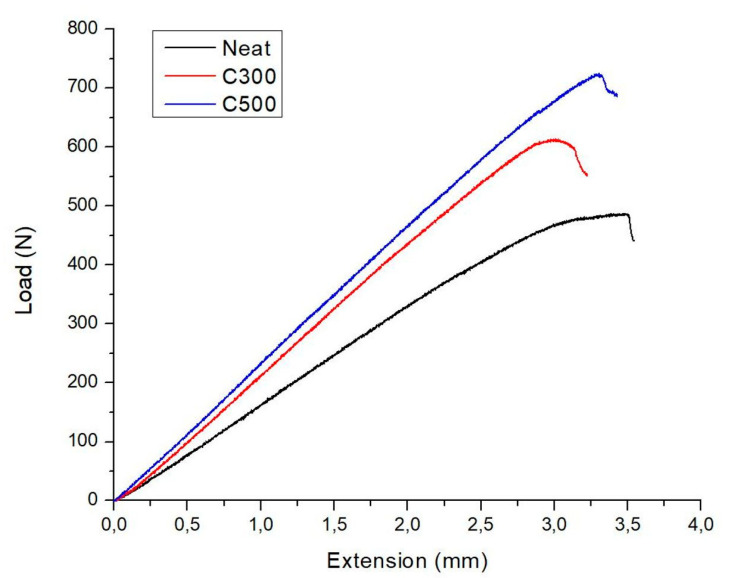
Load-displacement curves of representative specimens under ENF tests.

**Figure 7 polymers-12-03060-f007:**
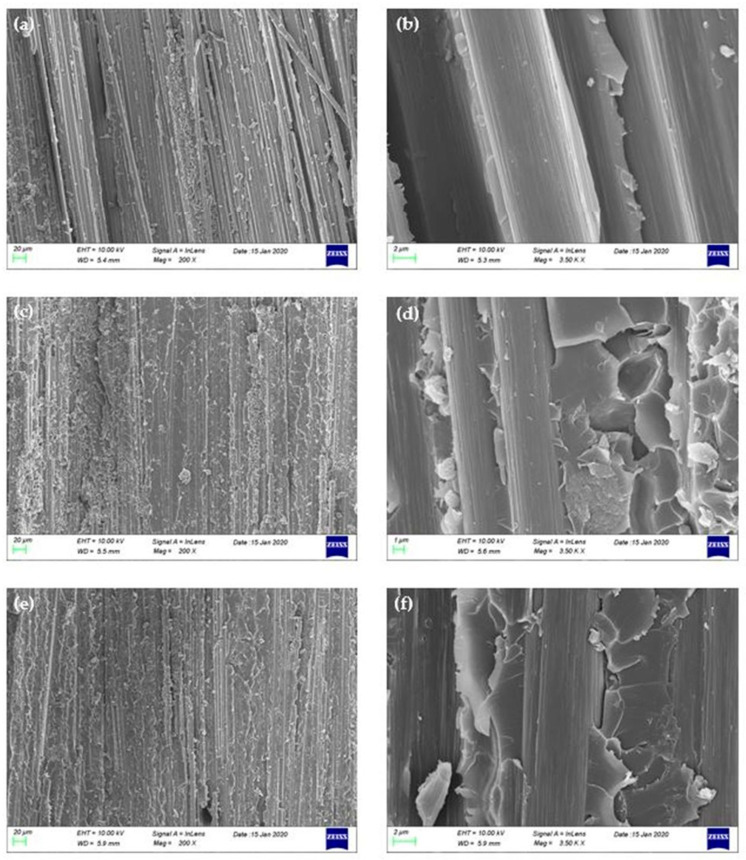
SEM images of representative fracture surfaces of, (**a**,**b**) neat composite, (**c**,**d**) 0.5 wt% C300 nano-modified composite, and (**e**,**f**) 0.5 wt% C500 nano-modified composite.

**Figure 8 polymers-12-03060-f008:**
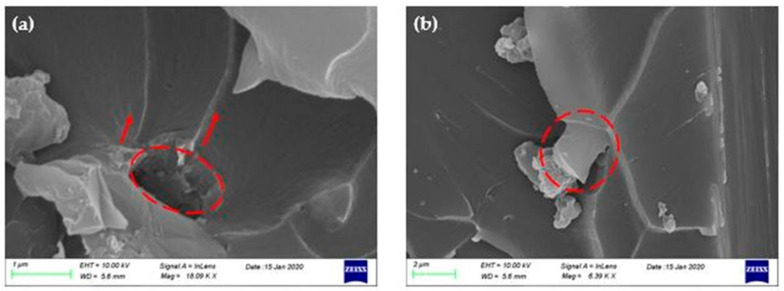
SEM images of C300 nano-fillers failure mechanisms: (**a**) crack pinning and bifurcation and (**b**) separation of graphene sheets and pull-out.

**Figure 9 polymers-12-03060-f009:**
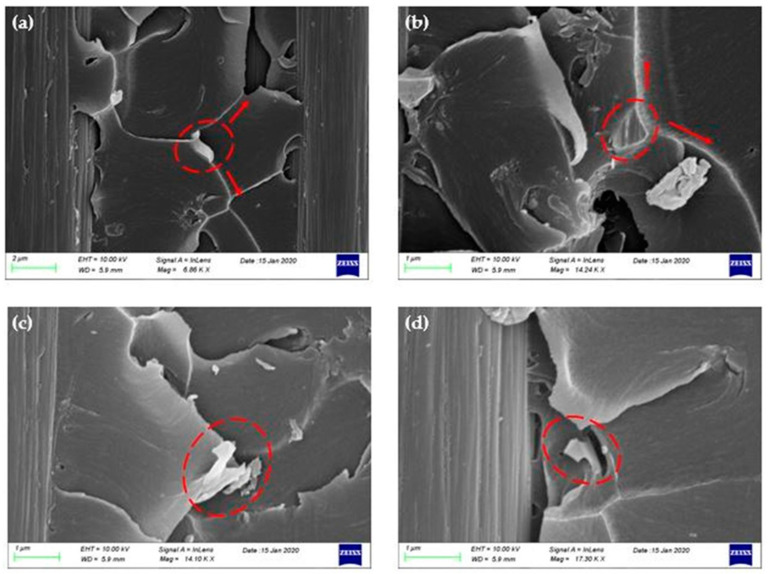
SEM images of C500 nano-fillers failure mechanisms: (**a**,**b**) crack pinning and bifurcation and (**c**,**d**) separation of graphene sheets and pull-out.

**Figure 10 polymers-12-03060-f010:**
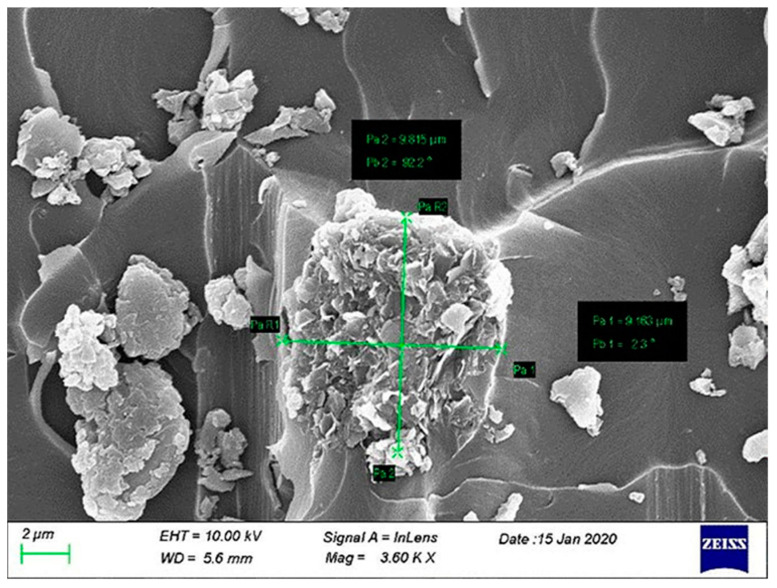
SEM image of C300 nano-fillers agglomerate 9.5 μm in size.

**Table 1 polymers-12-03060-t001:** Typical characteristics of components of the matrix system.

	Araldite LY 1556	Aradur 1571	Accelerator 1573	Hardener XB 3403
Component	Epoxy resin	Hardener paste	Accelerator paste	Hardener based on polyamine
Aspect (visual)	Clear, pale yellow liquid	White viscous paste	White viscous paste	Clear liquid
Viscosity at 25 °C (mPa·s)	10,000–12,000	28,000–40,000	60,000–90,000	5–20
Density at 25 °C (g/cm^3^)	1.15–1.2	1.2	1.08	1
Storage Temperature (°C)	2–40	<8	<8	2–40

**Table 2 polymers-12-03060-t002:** Physical characteristics of the used nano-fillers.

	xGNPs C300	xGNPs C500
Appearance	Black granules/powder	Black granules/powder
Density (g/cm^3^)	0.2–0.4	0.2–0.4
Specific Surface Area (m^2^/g)	300	500
Diameter (μm)	<2	<2
Typical Particle Thickness	Few nanometers	Few nanometers

**Table 3 polymers-12-03060-t003:** Fiber volume faction of manufactured composites.

Material	Fiber Volume Fraction
Neat	61%
C300	58%
C500	59%

**Table 4 polymers-12-03060-t004:** Interlaminar fracture toughness, G_IC_, of manufactured composites.

Material	Interlaminar Fracture Toughness G_IC_ (kJ/m^2^)	
	AVG	STDEV
Neat	0.55	0.079
0.5 wt% C300	0.48	0.061
0.5 wt% C500	0.69	0.084

**Table 5 polymers-12-03060-t005:** Interlaminar fracture toughness, G_IIC_, of manufactured composites.

Material	Interlaminar Fracture Toughness G_IIC_ (kJ/m^2^)	
	AVG	STDEV
Neat	1.02	0.06
0.5 wt% C300	1.07	0.12
0.5 wt% C500	1.36	0.16

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
