# Peer review of "A Preliminary Study of the Influence of Graphene Nanoplatelet Specific Surface Area on the Interlaminar Fracture Properties of Carbon Fiber/Epoxy Composites"

_polymers, 2020, doi:10.3390/polym12123060_

Round 1

Reviewer 1 Report

In my opinion, the work presents a good scientific level, however some minor remarks highlighted below.

1. Fig 2. - The size of the photos showing the scans is too small, I suggest enlarging them a bit, you can also place illustrative photos of the prepared samples.

2. Fig 5. - The colors of the lines in the legend are hard to recognize

3. It would also be valuable to include the results of the measurements from the tensile or flexural tests, to confirm the properties improvement.

Author Response

Response to Reviewer 1 Comments

Point 1: 
Fig 2. - The size of the photos showing the scans is too small, I suggest enlarging them a bit, you can also place illustrative photos of the prepared samples.

Response 1: In page 6 the Figure 2 has changed. The photos of c-scans have been enlarged and illustrative photos of the manufactured laminates have been also added.

Point 2: Fig 5. - The colors of the lines in the legend are hard to recognize.

Response 2: In page 8 in Figure 5 the lines in the legend have been improved.

Point 3: It would also be valuable to include the results of the measurements from the tensile or flexural tests, to confirm the properties improvement.

Response 3: Tensile and flexural tests have been conducted only for the polymer and the results are represented in the following graphs. As it is observed, there is no significant enhancement in the tensile and flexural properties of the polymer matrix, that is why these results were not included in the main text.

Reviewer 2 Report

The paper titled “Influence of Graphene Nanoplatelet Specific Surface Area on the Fracture Properties of Carbon Fiber Composites” concerns to a very interesting topic either form an applied and scientific view points. In the concerning to the applied importance of carbon fiber epoxy reinforced parts doped by grapheme nanoplatelets, the work matches the objective by since a technical report perspective, but just so.

Moreover, from the title, the reader may interpret that an exhaustive study about the specific surface area of the nanoplatelets, but the authors just use two commercial ones at a same theoretical amount (0.5 w/w) but doesn´t offer further evidences that this is the real amount of GNP in the composite. In absence of this information, all the discussion may be highly spurious and non sense. PLEASE, check it and provide this information.

Additionally, the C-Scan results need more info. The images presented are hard to be understood by a lay man in absence of comments and details about. The observed differences are not enough, at a glance, as to being associated to the very little difference between the declared fiber volume fraction between C300 and C500 (mainly). There are other many concerns, but the ones are enough as to make a decision about.

In essence, the work is focused on just two types of GNPs in epoxy-carbon composites and the use of two conventional fracture tests to conclude that the one with higher specific surface (SSA) area throw some better properties but avoiding to explain the scientific reason of the latter. This reviewer has found that the authors think that the existence of a certain threshold SSA must exist. Well, in the opinion of this reviewer, this is the really interesting to be ascertained, but a so simply experimental design as the authors has employed is far of being able to identify this point. Just as a suggestion, the employ of DOEs would the authors to make a more complete and convenient investigation about.

Consequently, it the actual state, the article is a good technical report but never a scientific article. No seeds of scientific explanation can be found in the article. In the case that POLYMERS may accept technical reports, the article may be considered. In the case that POLYMERS is related to the polymer science field, my opinion is that the article must be rejected. Neither scientific nor technical novelties are robustly neither found nor presented.

In any case, this reviewer (by following his personal expertise in the area) opines that the article is much more appropriate to a mechanical engineering journal than for a polymer based journal, on the basis that very little about the polymer character of the composite is discussed under a scientific viewpoint.

Recommendation: REJECT.

Author Response

Response to Reviewer 2 Comments

Point 1: Moreover, from the title, the reader may interpret that an exhaustive study about the specific surface area of the nanoplatelets, but the authors just use two commercial ones at a same theoretical amount (0.5 w/w) but doesn´t offer further evidences that this is the real amount of GNP in the composite. In absence of this information, all the discussion may be highly spurious and non sense. PLEASE, check it and provide this information.

Response 1: First of all, the title of the paper doesn’t mention that an exhaustive study about the SSA of the graphene nanoplatelets would be carried out. Moreover, the exact types of GNPs used are cited in the abstract (line 20), so none reader can interpret something different.

Secondly, the reasons, why one weight fraction of 0.5% wt was selected, were:

  1. In order to avoid agglomerations at higher weight contents, which act as defects and reduce the performance of CFRPs.
  2. For comparison reasons with other graphene nano-species (GNSs). The same weight fraction of 0.5% wt GNSs was used also in previous study in our laboratory, in order to evaluate the fracture toughness of the nano-doped composites with GNSs.

Point 2: Additionally, the C-Scan results need more info. The images presented are hard to be understood by a lay man in absence of comments and details about. The observed differences are not enough, at a glance, as to being associated to the very little difference between the declared fiber volume fraction between C300 and C500 (mainly). There are other many concerns, but the ones are enough as to make a decision about.

Response 2: In page 5, line 186, more info about the C-Scan results have been added.

Point 3: In essence, the work is focused on just two types of GNPs in epoxy-carbon composites and the use of two conventional fracture tests to conclude that the one with higher specific surface (SSA) area throw some better properties but avoiding to explain the scientific reason of the latter. This reviewer has found that the authors think that the existence of a certain threshold SSA must exist. Well, in the opinion of this reviewer, this is the really interesting to be ascertained, but a so simply experimental design as the authors has employed is far of being able to identify this point. Just as a suggestion, the employ of DOEs would the authors to make a more complete and convenient investigation about.

Response 3: The scientific reason for the conclusions is explained analytically in Section 4.4.

Furthermore, the existence of a threshold in SSA is clearly mentioned in abstract (line 31,32), but also in page 8, lines 249,250 and 254.

Point 4: Consequently, it the actual state, the article is a good technical report but never a scientific article. No seeds of scientific explanation can be found in the article. In the case that POLYMERS may accept technical reports, the article may be considered. In the case that POLYMERS is related to the polymer science field, my opinion is that the article must be rejected. Neither scientific nor technical novelties are robustly neither found nor presented.

Response 4: As it was referred in the above response (3), there is scientific explanation in the article.

Point 5: In any case, this reviewer (by following his personal expertise in the area) opines that the article is much more appropriate to a mechanical engineering journal than for a polymer based journal, on the basis that very little about the polymer character of the composite is discussed under a scientific viewpoint.

Response 5: The article belongs to the Topical Collection "Reinforced Polymer Composites". As editor mention, the Collection focuses on how reinforcing fillers can improve the polymer properties. By adding the GNPs into the polymer matrix and conducting the fracture tests (mode I and II), the aim was to highlight the absorption mechanisms of GNPs either into the matrix (polymer) or in the matrix-fiber interphase. The corresponding SEM images and the interaction between the polymer-additives are extensively cited in Section 4.3.

Reviewer 3 Report

The paper investigates the effect of the specific surface area of GNPs on the fracture surface of CFRPs. The paper is well written but, in my opinion, it is not suitable for Polymers journal in the present form since the topic focused on the improvement of fracture surface by GNPs is more suitable for other MDPI journals, such as Materials for example.

The characterization of nano-modified matrix and the dispersion of GNPs is missing. It is an important issue issue for the readers of Polymers journal but it is also very important to correlate the mechanical results with the state of dispersion of GNPs inside the matrix.

I suggest to explain more clearly the reasons of the different state of dispersion between C300 and C500 fillers in the epoxy matrix and to correlate it with the specific surface area of the two types of GNPs particles.

Finally, I suggest completing the state of the art by citing some recently published works on the subjects. See for example: 1) Polymers 2020, 12(9), 1895; https://doi.org/10.3390/polym12091895; 2) Front. Mater., 10 July 2019 | https://doi.org/10.3389/fmats.2019.00156; 3) Composites Part B. 114, 175–183. doi: 10.1016/j.compositesb.2017.01.032; 4) Mater. Res. Express 6, 2053–2071. doi: 10.1088/2053-1591/aaeaf0.

Author Response

Response to Reviewer 3 Comments

Point 1: The paper investigates the effect of the specific surface area of GNPs on the fracture surface of CFRPs. The paper is well written but, in my opinion, it is not suitable for Polymers journal in the present form since the topic focused on the improvement of fracture surface by GNPs is more suitable for other MDPI journals, such as Materials for example.

Response 1: The article belongs to the Topical Collection "Reinforced Polymer Composites". As editor mention, the Collection focuses on how reinforcing fillers can improve the polymer properties. By adding the GNPs into the polymer matrix and conducting the fracture tests (mode I and II), the aim was to highlight the absorption mechanisms of GNPs either into the matrix (polymer) or in the matrix-fiber interphase.

Point 2: The characterization of nano-modified matrix and the dispersion of GNPs is missing. It is an important issue for the readers of Polymers journal but it is also very important to correlate the mechanical results with the state of dispersion of GNPs inside the matrix.

Response 2: The characterization of nano-modified matrix is analytically cited in Section 4.3. The SEM images represented focused on the absorption mechanisms of GNPs either into the matrix (polymer) or in the matrix-fiber interphase, as mentioned in previous response (1). Studying the mechanical properties and the state of dispersion of GNPs inside the matrix was out of the scope of the present study.

Point 3: I suggest to explain more clearly the reasons of the different state of dispersion between C300 and C500 fillers in the epoxy matrix and to correlate it with the specific surface area of the two types of GNPs particles.

Response 3: The different state of dispersion between C300 and C500 fillers in the epoxy matrix and its correlation with the specific surface area of the two types of GNPs particles have been clarified in Section 4.4.

Point 4: Finally, I suggest completing the state of the art by citing some recently published works on the subjects. See for example: 1) Polymers 2020, 12(9), 1895; https://doi.org/10.3390/polym12091895; 2) Front. Mater., 10 July 2019 | https://doi.org/10.3389/fmats.2019.00156; 3) Composites Part B. 114, 175–183. doi: 10.1016/j.compositesb.2017.01.032; 4) Mater. Res. Express 6, 2053–2071. doi: 10.1088/2053-1591/aaeaf0.

Response 4: Some of the above-mentioned articles have been added in the corresponding spots in the text (lines 45 and 428-437).

Round 2

Reviewer 2 Report

See attached report for color details.

The authors have partially answered to my concerns, but in general without solving the main criticisms of the previous revision draft. The reviewer´s comments about are after the ones by the authors, and has been written in blue. In essence, this reviewer must stay in the previous recommendation of REJECTING the article:

Point 1: Moreover, from the title, the reader may interpret that an exhaustive study about the specific surface area of the nanoplatelets, but the authors just use two commercial ones at a same theoretical amount (0.5 w/w) but doesn´t offer further evidences that this is the real amount of GNP in the composite. In absence of this information, all the discussion may be highly spurious and non sense. PLEASE, check it and provide this information.

Response 1: First of all, the title of the paper doesn’t mention that an exhaustive study about the SSA of the graphene nanoplatelets would be carried out. Moreover, the exact types of GNPs used are cited in the abstract (line 20), so none reader can interpret something different.

Secondly, the reasons, why one weight fraction of 0.5% wt was selected, were:

  1. In order to avoid agglomerations at higher weight contents, which act as defects and reduce the performance of CFRPs.
  2. For comparison reasons with other graphene nano-species (GNSs). The same weight fraction of 0.5% wt GNSs was used also in previous study in our laboratory, in order to evaluate the fracture toughness of the nano-doped composites with GNSs.

Reviewer Answer 1:

First of all, although the paper title doesn´t indicate that it is an exhaustive study, this is so general that one may think that the purpose of the article is also more general, and not a single comparison of two different commercial GNPs. Note that your article is titled as “Influence of Graphene Nanoplatelet Specific Surface area on the Fracture Properties of Carbon Fiber Composites”, and this reviewer, as a simple reader, really interpret that the authors have really study this influence in detail, don´t you? In fact, this is so general that in the title the authors not even identify the type of composite, that means, if thermoset or thermoplastic based composites. Nevertheless, the authors must consider that the influence of the specific surface of the GNPs cannot be ascertained in a robust manner by the simply comparison of two different ones, but a third (at least) must be included to observe that this is true or not. Please, reconsider it for further works.

The reasons given for using just 0.5% may be accepted. But since the authors say that they at acts as defect, this reviewer wonders why they have not used even lower amounts. And even more when they recognize that agglomerations are produced (so, they are unable to avoid them). Well, in the opinion of this reviewer this fact is a source of additional uncertainty on the measurements as to robustly conclude that it is the specific area the cause of the observed changes.

Additionally, in my previous revision draft this reviewer asked the authors about the real content and not just the nominal (…but the authors just use two commercial ones at a same theoretical amount (0.5 w/w) but doesn´t offer further evidences that this is the real,…) and they have avoided to answer to this PRIME question. Is there any reason for that? In absence of this information, whatever discussion becomes spurious.

Point 2: Additionally, the C-Scan results need more info. The images presented are hard to be understood by a lay man in absence of comments and details about. The observed differences are not enough, at a glance, as to being associated to the very little difference between the declared fiber volume fraction between C300 and C500 (mainly). There are other many concerns, but the ones are enough as to make a decision about.

Response 2: In page 5, line 186, more info about the C-Scan results have been added.

Reviewer Answer 5:

Ok. This is now been solved, and the reader is able to ascertain which of the composites has more defects.

Point 3: In essence, the work is focused on just two types of GNPs in epoxy-carbon composites and the use of two conventional fracture tests to conclude that the one with higher specific surface (SSA) area throw some better properties but avoiding to explain the scientific reason of the latter. This reviewer has found that the authors think that the existence of a certain threshold SSA must exist. Well, in the opinion of this reviewer, this is the really interesting to be ascertained, but a so simply experimental design as the authors has employed is far of being able to identify this point. Just as a suggestion, the employ of DOEs would the authors to make a more complete and convenient investigation about.

Response 3: The scientific reason for the conclusions is explained analytically in Section 4.4.

Furthermore, the existence of a threshold in SSA is clearly mentioned in abstract (line 31,32), but also in page 8, lines 249,250 and 254.

Reviewer Answer 5:

Probably this reviewer was unable to transmit to the authors that my query was in the sense that, although the authors think there is a threshold in the SSA, the comparison of two of them not enough as to conclude this fact. Note that this reviewer, as researched, knows and believe in the existence of this threshold. But the knowledge of this fact is not enough as to be published unless this is robustly demonstrated, and the comparison of mere two different SSA is far of being enough since a scientific viewpoint. Just so.

Point 4: Consequently, it the actual state, the article is a good technical report but never a scientific article. No seeds of scientific explanation can be found in the article. In the case that POLYMERS may accept technical reports, the article may be considered. In the case that POLYMERS is related to the polymer science field, my opinion is that the article must be rejected. Neither scientific nor technical novelties are robustly neither found nor presented.

Response 4: As it was referred in the above response (3), there is scientific explanation in the article.

Reviewer Answer 5:

Ok. There is a quite good scientific explanation based on other works in literature. But the fact is that the authors use these to conclude about two simple SSA without checking with a control one (at least another one SSA) is mandatorily required. If not, you simply compare two ideal cases, clearly insufficient as to conclude about the influence of the specific area in a general and predicting manner.

Point 5: In any case, this reviewer (by following his personal expertise in the area) opines that the article is much more appropriate to a mechanical engineering journal than for a polymer based journal, on the basis that very little about the polymer character of the composite is discussed under a scientific viewpoint.

Response 5: The article belongs to the Topical Collection "Reinforced Polymer Composites". As editor mention, the Collection focuses on how reinforcing fillers can improve the polymer properties. By adding the GNPs into the polymer matrix and conducting the fracture tests (mode I and II), the aim was to highlight the absorption mechanisms of GNPs either into the matrix (polymer) or in the matrix-fiber interphase. The corresponding SEM images and the interaction between the polymer-additives are extensively cited in Section 4.3.

Reviewer Answer 5:

This reviewer has no doubt that the topic of the article is focused on Polymer Reinforcement Composites. This is clear, the topic matches the goal, but just so. What is not so evident is if the type of study performed matches the requirements of a polymer science based journal. There are plenty of factor influencing the properties the authors have studied, and one of the most important and influencing is the real amount of the GNP present in the tested champions, and none of this has been explained in the text. In my own experience, one think is the nominal amount of filler used, and other very different the real amount of it in the final champion to be tested. Usually, there are many differences between these two values depending of the quality on the dispersion and the processing steps. In this work, surprisingly, the authors have avoided to answer this basic question in the Point 1. Note that a tiny difference in these values may explain, by far, all the results that the authors have discussed, and consequently, the whole conclusions become spurious.

With the above mentioned comments, this reviewer must stay in the previous draft recommendation: REJECT.

Reviewer 3 Report

The authors answered all the comments.

Author Response

Thank you for approving our responses